# Reconstructing perceived faces from brain activations with deep adversarial neural decoding

**Yağmur Güçlütürk\*, Umut Güçlü\*,**
**Katja Seeliger, Sander Bosch,**
**Rob van Lier, Marcel van Gerven,**
Radboud University, Donders Institute for Brain, Cognition and Behaviour
Nijmegen, the Netherlands
{y.gucluturk, u.guclu}@donders.ru.nl

## Abstract

Here, we present a novel approach to solve the problem of reconstructing perceived stimuli from brain responses by combining probabilistic inference with deep learning. Our approach first inverts the linear transformation from latent features to brain responses with maximum a posteriori estimation and then inverts the nonlinear transformation from perceived stimuli to latent features with adversarial training of convolutional neural networks. We test our approach with a functional magnetic resonance imaging experiment and show that it can generate state-of-the-art reconstructions of perceived faces from brain activations.

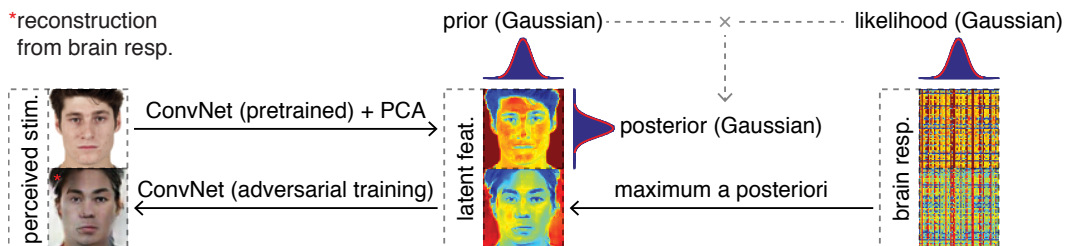

Figure 1: An illustration of our approach to solve the problem of reconstructing perceived stimuli from brain responses by combining probabilistic inference with deep learning.

## 1 Introduction

A key objective in sensory neuroscience is to characterize the relationship between perceived stimuli and brain responses. This relationship can be studied with neural encoding and neural decoding in functional magnetic resonance imaging (fMRI) [1]. The goal of neural encoding is to predict brain responses to perceived stimuli [2]. Conversely, the goal of neural decoding is to classify [3, 4], identify [5, 6] or reconstruct [7–11] perceived stimuli from brain responses.

The recent integration of deep learning into neural encoding has been a very successful endeavor [12, 13]. To date, the most accurate predictions of brain responses to perceived stimuli have been achieved with convolutional neural networks [14–20], leading to novel insights about the functional organization of neural representations. At the same time, the use of deep learning as the basis for neural decoding has received less widespread attention. Deep neural networks have been used for classifying or identifying stimuli via the use of a deep encoding model [16, 21] or by predicting

intermediate stimulus features [22, 23]. Deep belief networks and convolutional neural networks have been used to reconstruct basic stimuli (handwritten characters and geometric figures) from patterns of brain activity [24, 25]. To date, going beyond such mostly retinotopy-driven reconstructions and reconstructing complex naturalistic stimuli with high accuracy have proven to be difficult.

The integration of deep learning into neural decoding is an exciting approach for solving the reconstruction problem, which is defined as the inversion of the (non)linear transformation from perceived stimuli to brain responses to obtain a reconstruction of the original stimulus from patterns of brain activity alone. Reconstruction can be formulated as an inference problem, which can be solved by maximum a posteriori estimation. Multiple variants of this formulation have been proposed in the literature [26–30]. At the same time, significant improvements are to be expected from deep neural decoding given the success of deep learning in solving image reconstruction problems in computer vision such as colorization [31], face hallucination [32], inpainting [33] and super-resolution [34].

Here, we present a new approach by combining probabilistic inference with deep learning, which we refer to as deep adversarial neural decoding (DAND). Our approach first inverts the linear transformation from latent features to observed responses with maximum a posteriori estimation. Next, it inverts the nonlinear transformation from perceived stimuli to latent features with adversarial training and convolutional neural networks. An illustration of our model is provided in Figure 1. We show that our approach achieves state-of-the-art reconstructions of perceived faces from the human brain.

## 2 Methods

### 2.1 Problem statement

Let $\mathbf{x} \in \mathbb{R}^{h \times w \times c}$, $\mathbf{z} \in \mathbb{R}^{p}$, $\mathbf{y} \in \mathbb{R}^{q}$ be a stimulus, feature, response triplet, and $\phi : \mathbb{R}^{h \times w \times c} \to \mathbb{R}^{p}$ be a latent feature model such that $\mathbf{z} = \phi(\mathbf{x})$ and $\mathbf{x} = \phi^{-1}(\mathbf{z})$. Without loss of generality, we assume that all of the variables are normalized to have zero mean and unit variance.

We are interested in solving the problem of reconstructing perceived stimuli from brain responses:

$$\hat{\mathbf{x}} = \phi^{-1}(\arg \max_{\mathbf{z}} \Pr(\mathbf{z} \mid \mathbf{y})) \tag{1}$$

where $\Pr(\mathbf{z} \mid \mathbf{y})$ is the posterior. We reformulate the posterior through Bayes' theorem:

$$\hat{\mathbf{x}} = \phi^{-1}\left(\arg \max_{\mathbf{z}} \left[\Pr(\mathbf{y} \mid \mathbf{z}) \Pr(\mathbf{z})\right]\right) \tag{2}$$

where $\Pr(\mathbf{y} \mid \mathbf{z})$ is the likelihood, and $\Pr(\mathbf{z})$ is the prior. In the following subsections, we define the latent feature model, the likelihood and the prior.

### 2.2 Latent feature model

We define the latent feature model $\phi(\mathbf{x})$ by modifying the VGG-Face pretrained model [35]. This model is a 16-layer convolutional neural network, which was trained for face recognition. First, we truncate it by retaining the first 14 layers and discarding the last two layers of the model. At this point, the truncated model outputs 4096-dimensional latent features. To reduce the dimensionality of the latent features, we then combine the model with principal component analysis by estimating the loadings that project the 4096-dimensional latent features to the first 699 principal component scores (maximum number of components given the number of training observations) and adding them at the end of the truncated model as a new fully-connected layer. At this point, the combined model outputs 699-dimensional latent features.

Following the ideas presented in [36–38], we define the inverse of the feature model $\phi^{-1}(\mathbf{z})$ (i.e., the image generator) as a convolutional neural network which transforms the 699-dimensional latent variables to $64 \times 64 \times 3$ images and estimate its parameters via an adversarial process. The generator comprises five deconvolution layers: The $i$th layer has $2^{10-i}$ kernels with a size of $4 \times 4$, a stride of $2 \times 2$, a padding of $1 \times 1$, batch normalization and rectified linear units. Exceptions are the first layer which has a stride of $1 \times 1$, and no padding; and the last layer which has three kernels, no batch normalization [39] and hyperbolic tangent units. Note that we do use the inverse of the loadings in the generator.

To enable adversarial training, we define a discriminator ($\psi$) along with the generator. The discriminator comprises five convolution layers. The $i$th layer has $2^{5+i}$ kernels with a size of $4 \times 4$, a stride of $2 \times 2$, a padding of $1 \times 1$, batch normalization and leaky rectified linear units with a slope of $0.2$ except for the first layer which has no batch normalization and last layer which has one kernel, a stride of $1 \times 1$, no padding, no batch normalization and a sigmoid unit.

We train the generator and the discriminator by pitting them against each other in a two-player zero-sum game, where the goal of the discriminator is to discriminate stimuli from reconstructions and the goal of the generator is to generate reconstructions that are indiscriminable from original stimuli. This ensures that reconstructed stimuli are similar to target stimuli on a pixel level and a feature level.

The discriminator is trained by iteratively minimizing the following discriminator loss function:

$$L_{\text{dis}} = -\mathbb{E}\left[\log(\psi(\mathbf{x})) + \log(1 - \psi(\phi^{-1}(\mathbf{z})))\right] \tag{3}$$

where $\psi$ is the output of the discriminator which gives the probability that its input is an original stimulus and not a reconstructed stimulus. The generator is trained by iteratively minimizing a generator loss function, which is a linear combination of an adversarial loss function, a feature loss function and a stimulus loss function:

$$L_{\text{gen}} = -\lambda_{\text{adv}} \underbrace{\mathbb{E}\left[\log(\psi(\phi^{-1}(\mathbf{z})))\right]}_{L_{\text{adv}}} + \lambda_{\text{fea}} \underbrace{\mathbb{E}[\|\xi(\mathbf{x}) - \xi(\phi^{-1}(\mathbf{z}))\|^2]}_{L_{\text{fea}}} + \lambda_{\text{sti}} \underbrace{\mathbb{E}[\|\mathbf{x} - \phi^{-1}(\mathbf{z})\|^2]}_{L_{\text{sti}}} \tag{4}$$

where $\xi$ is the relu3_3 outputs of the pretrained VGG-16 model [40, 41]. Note that the targets and the reconstructions are lower resolution (i.e., $64 \times 64$) than the images that are used to obtain the latent features (i.e., $224 \times 224$).

## 2.3 Likelihood and prior

We define the likelihood as a multivariate Gaussian distribution over $\mathbf{y}$:

$$\Pr(\mathbf{y}|\mathbf{z}) = \mathcal{N}_{\mathbf{y}}(\mathbf{B}^\top \mathbf{z}, \boldsymbol{\Sigma}) \tag{5}$$

where $\mathbf{B} = (\boldsymbol{\beta}_1, \ldots, \boldsymbol{\beta}_q) \in \mathbb{R}^{p \times q}$ and $\boldsymbol{\Sigma} = \text{diag}(\sigma_1^2, \ldots, \sigma_q^2) \in \mathbb{R}^{q \times q}$. Here, the features $\times$ voxels matrix $\mathbf{B}$ contains the learnable parameters of the likelihood in its columns $\boldsymbol{\beta}_i$ (which can also be interpreted as regression coefficients of a linear regression model, which predicts $\mathbf{y}$ from $\mathbf{z}$).

We estimate the parameters with ordinary least squares, such that $\hat{\boldsymbol{\beta}}_i = \arg\min_{\boldsymbol{\beta}_i} \mathbb{E}[\|y_i - \boldsymbol{\beta}_i^\top \mathbf{z}\|^2]$ and $\hat{\sigma}_i^2 = \mathbb{E}[\|y_i - \hat{\boldsymbol{\beta}}_i^\top \mathbf{z}\|^2]$.

We define the prior as a zero mean and unit variance multivariate Gaussian distribution $\Pr(\mathbf{z}) = \mathcal{N}_{\mathbf{z}}(\mathbf{0}, \mathbf{I})$.

## 2.4 Posterior

To derive the posterior (2), we first reformulate the likelihood as a multivariate Gaussian distribution over $\mathbf{z}$. That is, after taking out constant terms with respect to $\mathbf{z}$ from the likelihood, it immediately becomes proportional to the canonical form Gaussian over $\mathbf{z}$ with $\nu = \mathbf{B}\boldsymbol{\Sigma}^{-1}\mathbf{y}$ and $\Lambda = \mathbf{B}\boldsymbol{\Sigma}^{-1}\mathbf{B}^\top$, which is equivalent to the standard form Gaussian with mean $\Lambda^{-1}\nu$ and covariance $\Lambda^{-1}$.

This allows us to write:

$$\Pr(\mathbf{z}|\mathbf{y}) \propto \mathcal{N}_{\mathbf{z}}\left(\Lambda^{-1}\nu, \Lambda^{-1}\right)\mathcal{N}_{\mathbf{z}}(\mathbf{0}, \mathbf{I}) \tag{6}$$

Next, recall that the product of two multivariate Gaussians can be formulated in terms of one multivariate Gaussian [42]. That is, $\mathcal{N}_{\mathbf{z}}(\mathbf{m}_1, \boldsymbol{\Sigma}_1)\mathcal{N}_{\mathbf{z}}(\mathbf{m}_2, \boldsymbol{\Sigma}_2) \propto \mathcal{N}_{\mathbf{z}}(\mathbf{m}_c, \boldsymbol{\Sigma}_c)$ with $\mathbf{m}_c = \left(\boldsymbol{\Sigma}_1^{-1} + \boldsymbol{\Sigma}_2^{-1}\right)^{-1}\left(\boldsymbol{\Sigma}^{-1}\mathbf{m}_1 + \boldsymbol{\Sigma}_2^{-1}\mathbf{m}_2\right)$ and $\boldsymbol{\Sigma}_c = \left(\boldsymbol{\Sigma}_1^{-1} + \boldsymbol{\Sigma}_2^{-1}\right)^{-1}$. By plugging this formulation into Equation (6), we obtain $\Pr(\mathbf{z}|\mathbf{y}) \propto \mathcal{N}_{\mathbf{z}}(\mathbf{m}_c, \boldsymbol{\Sigma}_c)$ with $\mathbf{m}_c = (\mathbf{B}\boldsymbol{\Sigma}^{-1}\mathbf{B}^\top + \mathbf{I})^{-1}\mathbf{B}\boldsymbol{\Sigma}^{-1}\mathbf{y}$ and $\boldsymbol{\Sigma}_c = (\mathbf{B}\boldsymbol{\Sigma}^{-1}\mathbf{B}^\top + \mathbf{I})^{-1}$.

Recall that we are interested in reconstructing stimuli from responses by generating reconstructions from the features that maximize the posterior. Notice that the (unnormalized) posterior is maximized

at its mean $\mathbf{m}_c$ since this corresponds to the mode for a multivariate Gaussian distribution. Therefore, the solution of the problem of reconstructing stimuli from responses reduces to the following simple expression:

$$\hat{\mathbf{x}} = \phi^{-1}\left((\mathbf{B}\boldsymbol{\Sigma}^{-1}\mathbf{B}^{\top} + \mathbf{I})^{-1}\mathbf{B}\boldsymbol{\Sigma}^{-1}\mathbf{y}\right) \tag{7}$$

## 3 Results

### 3.1 Datasets

We used the following datasets in our experiments:

**fMRI dataset**. We collected a new fMRI dataset, which comprises face stimuli and associated blood-oxygen-level dependent (BOLD) responses. The stimuli used in the fMRI experiment were drawn from [43–45] and other online sources, and consisted of photographs of front-facing individuals with neutral expressions. We measured BOLD responses (TR = 1.4 s, voxel size = $2 \times 2 \times 2$ mm$^3$, whole-brain coverage) of two healthy adult subjects (S1: 28-year old female; S2: 39-year old male) as they were fixating on a target ($0.6 \times 0.6$ degree) [46] superimposed on the stimuli ($15 \times 15$ degrees). Each face was presented at 5 Hz for 1.4 s and followed by a middle gray background presented for 2.8 s. In total, 700 faces were presented twice for the training set, and 48 faces were repeated 13 times for the test set. The test set was balanced in terms of gender and ethnicity (based on the norming data provided in the original datasets). The experiment was approved by the local ethics committee (CMO Regio Arnhem-Nijmegen) and the subjects provided written informed consent in accordance with the Declaration of Helsinki. Our fMRI dataset is available from the first authors on reasonable request.

The stimuli were preprocessed as follows: Each image was cropped and resized to $224 \times 224$ pixels. This procedure was organized such that the distance between the top of the image and the vertical center of the eyes was 87 pixels, the distance between the vertical center of the eyes and the vertical center of the mouth was 75 pixels, the distance between the vertical center of the mouth and the bottom of the image was 61 pixels, and the horizontal center of the eyes and the mouth was at the horizontal center of the image.

The fMRI data were preprocessed as follows: Functional scans were realigned to the first functional scan and the mean functional scan, respectively. Realigned functional scans were slice time corrected. Anatomical scans were coregistered to the mean functional scan. Brains were extracted from the coregistered anatomical scans. Finally, stimulus-specific responses were deconvolved from the realigned and slice time corrected functional scans with a general linear model [47]. Here, deconvolution refers to estimating regression coefficients ($\mathbf{y}$) of the following GLMs: $\mathbf{y}^* = \mathbf{Xy}$, where $\mathbf{y}^*$ is raw voxel responses, $\mathbf{X}$ is HRF-convolved design matrix (one regressor per stimulus indicating its presence), and $\mathbf{y}$ is deconvolved voxel responses such that $\mathbf{y}$ is a vector of size $m \times 1$ with m denoting the number of unique stimuli, and there is one $\mathbf{y}$ per voxel.

**CelebA dataset** [48]. This dataset comprises 202599 in-the-wild portraits of 10177 people, which were drawn from online sources. The portraits are annotated with 40 attributes and five landmarks. We preprocessed the portraits as we preprocessed the stimuli in our fMRI dataset.

### 3.2 Implementation details

Our implementation makes use of Chainer and Cupy with CUDA and cuDNN [49] except for the following: The VGG-16 and VGG-Face pretrained models were ported to Chainer from Caffe [50]. Principal component analysis was implemented in scikit-learn [51]. fMRI preprocessing was implemented in SPM [52]. Brain extraction was implemented in FSL [53].

We trained the discriminator and the generator on the entire CelebA dataset by iteratively minimizing the discriminator loss function and the generator loss function in sequence for 100 epochs with Adam [54]. Model parameters were initialized as follows: biases were set to zero, the scaling parameters were drawn from $\mathcal{N}(\mathbf{1}, 2 \cdot 10^{-2}\mathbf{I})$, the shifting parameters were set to zero and the weights were drawn from $\mathcal{N}(\mathbf{1}, 10^{-2}\mathbf{I})$ [37]. We set the hyperparameters of the loss functions as follows: $\lambda_{\text{adv}} = 10^2$, $\lambda_{\text{dis}} = 10^2$, $\lambda_{\text{fea}} = 10^{-2}$ and $\lambda_{\text{sti}} = 2 \cdot 10^{-6}$ [38]. We set the hyperparameters of the optimizer as follows: $\alpha = 0.001$, $\beta_1 = 0.9$, $\beta_2 = 0.999$ and $\epsilon = 10^8$ [37].

We estimated the parameters of the likelihood term on the training split of our fMRI dataset.

### 3.3 Evaluation metrics

We evaluated our approach on the test split of our fMRI dataset with the following metrics: First, the feature similarity between the stimuli and their reconstructions, where the feature similarity is defined as the Euclidean similarity between the features, defined as the relu7 outputs of the VGG-Face pretrained model. Second, the Pearson correlation coefficient between the stimuli and their reconstructions. Third, the structural similarity between the stimuli and their reconstructions [55]. All evaluation was done on a held-out set not used at any point during model estimation or training. The voxels used in the reconstructions were selected as follows: For each test trial, $n$ voxels with smallest residuals (on training set) were selected. $n$ itself was selected such that reconstruction accuracy of remaining test trials was highest. We also performed an encoding analysis to see how well the latent features were predictive of voxel responses in different brain areas. The results of this analysis is reported in the supplementary material.

### 3.4 Reconstruction

We first demonstrate our results by reconstructing the stimulus images in the test set using i) the latent features and ii) the brain responses. Figure 2 shows 4 representative examples of the test stimuli and their reconstructions. The first column of both panels show the original test stimuli. The second column of both panels show the reconstructions of these stimuli $\mathbf{x}$ from the latent features $\mathbf{z}$ obtained by $\phi(\mathbf{x})$. These can be considered as an upper limit for the reconstruction accuracy of the brain responses since they are the best possible reconstructions that we can expect to achieve with a perfect neural decoder that can exactly predict the latent features from brain responses. The third and fourth columns of the figure show reconstructions of brain responses to stimuli of Subject 1 and Subject 2, respectively.

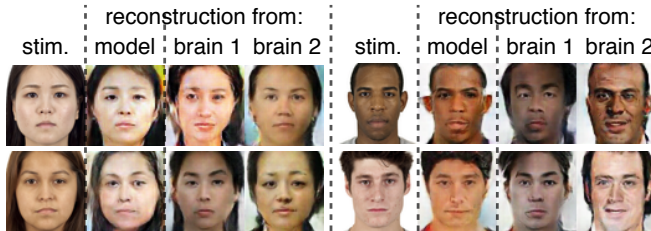

Figure 2: Reconstructions of the test stimuli from the latent features (model) and the brain responses of the two subjects (brain 1 and brain 2).

Visual inspection of the reconstructions from brain responses reveals that they match the test stimuli in several key aspects, such as gender, skin color and facial features. Table 1 shows the three reconstruction accuracy metrics for both subjects in terms of ratio of the reconstruction accuracy from brain responses to the reconstruction accuracy from latent features, which were significantly ($p < 0.05$, permutation test) above those for randomly sampled latent features (cf. 0.5181, 0.1532 and 0.5183, respectively).

Table 1: Reconstruction accuracy of the proposed decoding approach. The results are reported as the ratio of accuracy of reconstructing from brain responses and latent features.

|    | Feature similarity | Pearson correlation coefficient | Structural similarity |
|----|--------------------|----------------------------------|------------------------|
| S1 | $0.6546 \pm 0.0220$ | $0.6512 \pm 0.0493$ | $0.8365 \pm 0.0239$ |
| S2 | $0.6465 \pm 0.0222$ | $0.6580 \pm 0.0480$ | $0.8325 \pm 0.0229$ |

Furthermore, besides reconstruction accuracy, we tested the identification performance within and between groups that shared similar features (those that share gender or ethnicity as defined by the norming data were assumed to share similar features). Identification accuracies (which ranged between 57% and 62%) were significantly above chance-level (which ranged between 3% and 8%) in all cases ($p \ll 0.05$, Student's $t$-test). Furthermore, we found no significant differences between the identification accuracies when a reconstruction was identified among a group sharing similar features versus among a group that did not share similar features ($p > 0.79$, Student's $t$-test) (cf. [56]).

## 3.5 Visualization, interpolation and sampling

In the second experiment, we analyzed the properties of the stimulus features predictive of brain activations to characterize neural representations of faces. We first investigated the model representations to better understand what kind of features drive responses of the model. We visualized the features explaining the highest variance by independently setting the values of the first few latent dimensions to vary between their minimum and maximum values and generating reconstructions from these representations (Figure 3). As a result, we found that many of the latent features were coding for interpretable high level information such as age, gender, etc. For example, the first feature in Figure 3 appears to code for gender, the second one appears to code for hair color and complexion, the third one appears to code for age, and the fourth one appears to code for two different facial expressions.

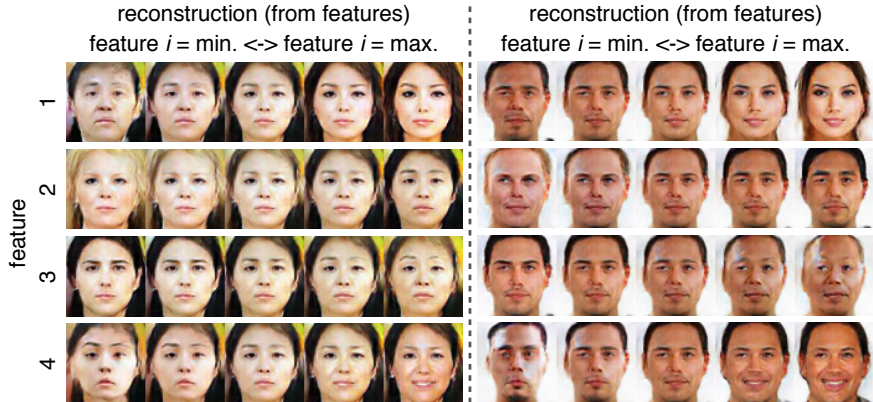

Figure 3: Reconstructions from features with single features set to vary between their minimum and maximum values.

We then explored the feature space that was learned by the latent feature model and the response space that was learned by the likelihood by systematically traversing the reconstructions obtained from different points in these spaces.

Figure 4A shows examples of reconstructions of stimuli from the latent features (rows one and four) and brain responses (rows two, three, five and six), as well as reconstructions from their interpolations between two points (columns three to nine). The reconstructions from the interpolations between two points show semantic changes with no sharp transitions.

Figure 4B shows reconstructions from latent features sampled from the model prior (first row) and from responses sampled from the response prior of each subject (second and third rows). The reconstructions from sampled representations are diverse and of high quality.

These results provide evidence that no memorization took place and the models learned relevant and interesting representations [37]. Furthermore, these results suggest that neural representations of faces might be embedded in a continuous and distributed space in the brain.

## 3.6 Comparison versus state-of-the-art

In this section we qualitatively (Figure 5) and quantitatively (Table 2) compare the performance of our approach with two existing decoding approaches from the literature*. Figure 5 shows example reconstructions from brain responses with three different approaches, namely with our approach, the eigenface approach [11, 57] and the identity transform approach [58, 29]. To achieve a fair comparison, the implementations of the three approaches only differed in terms of the feature models that were used, i.e. the eigenface approach had an eigenface (PCA) feature model and the identity transform approach had simply an identity transformation in place of the feature model.

Visual inspection of the reconstructions displayed in Figure 5 shows that DAND clearly outperforms the existing approaches. In particular, our reconstructions better capture the features of the stimuli

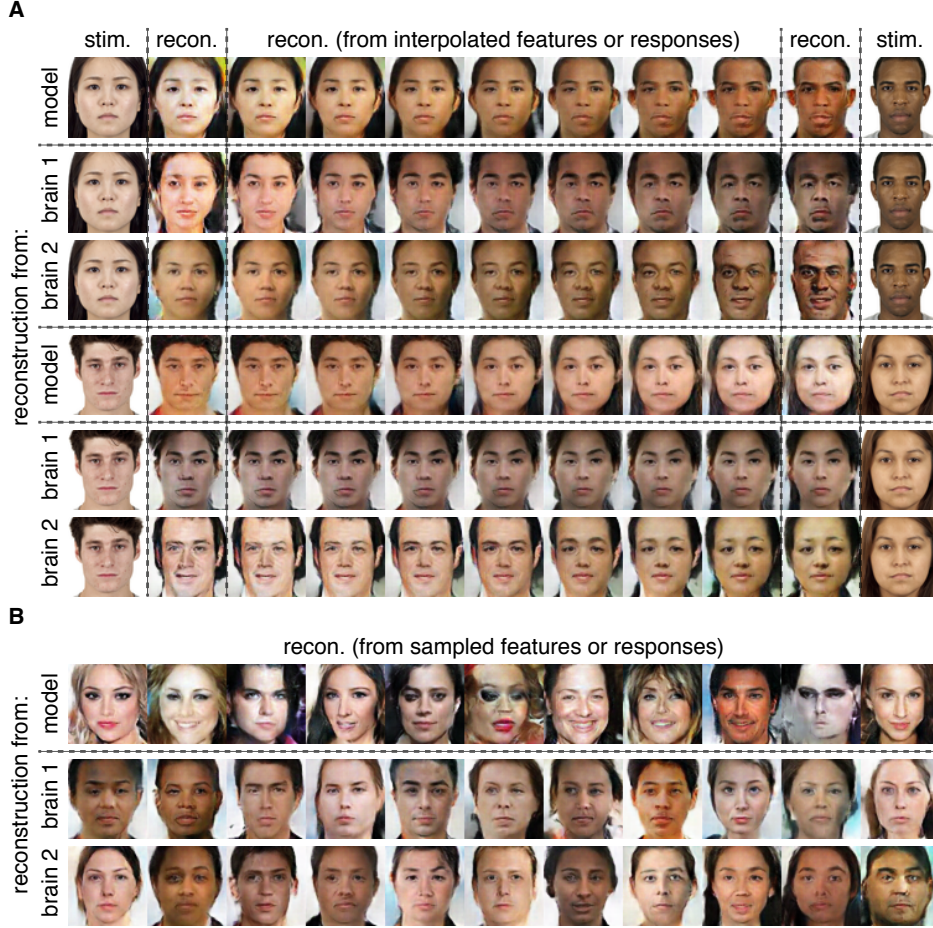

Figure 4: Reconstructions from interpolated (A) and sampled (B) latent features (model) and brain responses of the two subjects (brain 1 and brain 2).

such as gender, skin color and facial features. Furthermore, our reconstructions are more detailed, sharper, less noisy and more photorealistic than the eigenface and identity transform approaches. A quantitative comparison of the performance of the three approaches shows that the reconstruction accuracies achieved by our approach were significantly higher than those achieved by the existing approaches ($p \ll 0.05$, Student's $t$-test).

Table 2: Reconstruction accuracies of the three decoding approaches. LF denotes reconstructions from latent features.

|  |  | Feature similarity | Pearson correlation coefficient | Structural similarity |
|---|---|---|---|---|
| | S1 | $0.1254 \pm 0.0031$ | $0.4194 \pm 0.0347$ | $0.3744 \pm 0.0083$ |
| Identity | S2 | $0.1254 \pm 0.0038$ | $0.4299 \pm 0.0350$ | $0.3877 \pm 0.0083$ |
| | LF | $\mathbf{1.0000 \pm 0.0000}$ | $\mathbf{1.0000 \pm 0.0000}$ | $\mathbf{1.0000 \pm 0.0000}$ |
| | S1 | $0.1475 \pm 0.0043$ | $0.3779 \pm 0.0403$ | $0.3735 \pm 0.0102$ |
| Eigenface | S2 | $0.1457 \pm 0.0043$ | $0.2241 \pm 0.0435$ | $0.3671 \pm 0.0113$ |
| | LF | $0.3841 \pm 0.0149$ | $0.9875 \pm 0.0011$ | $0.9234 \pm 0.0040$ |
| | S1 | $\mathbf{0.1900 \pm 0.0052}$ | $\mathbf{0.4679 \pm 0.0358}$ | $\mathbf{0.4662 \pm 0.0126}$ |
| **DAND** | S2 | $\mathbf{0.1867 \pm 0.0054}$ | $\mathbf{0.4722 \pm 0.0344}$ | $\mathbf{0.4676 \pm 0.0130}$ |
| | LF | $0.2895 \pm 0.0137$ | $0.7181 \pm 0.0419$ | $0.5595 \pm 0.0181$ |

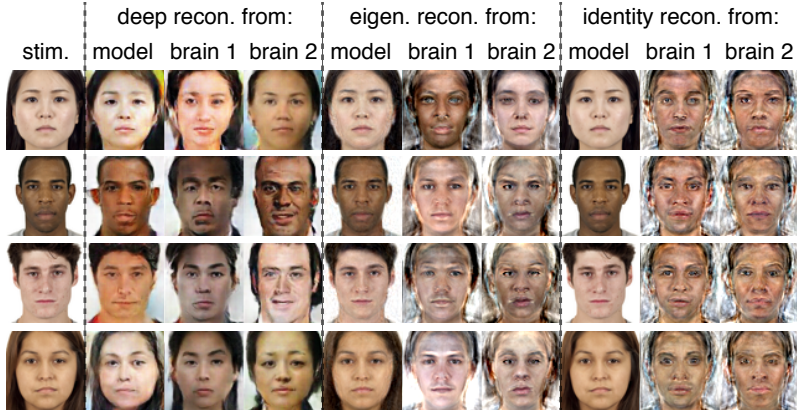

| | deep recon. from: | | | eigen. recon. from: | | | identity recon. from: | | |
| stim. | model | brain 1 | brain 2 | model | brain 1 | brain 2 | model | brain 1 | brain 2 |

Figure 5: Reconstructions from the latent features and brain responses of the two subjects (brain 1 and brain 2) using our decoding approach, as well as the eigenface and identity transform approaches for comparison.

### 3.7 Factors contributing to reconstruction accuracy

Finally, we investigated the factors contributing to the quality of reconstructions from brain responses. All of the faces in the test set had been annotated with 30 objective physical measures (such as nose width, face length, etc.) and 14 subjective measures (such as attractiveness, gender, ethnicity, etc.). Among these measures, we identified five subjective measures that are important for face perception [59–64] as measures of interest and supplemented them with an additional measure of stimulus complexity. Complexity was included because of its important role in visual perception [65]. The selected measures were attractiveness, complexity, ethnicity, femininity, masculinity and prototypicality. Note that the complexity measure was not part of the dataset annotations and was defined as the Kolmogorov complexity of the stimuli, which was taken to be their compressed file sizes [66].

To this end, we correlated the reconstruction accuracies of the 48 stimuli in the test set (for both subjects) with their corresponding measures (except for ethnicity) and used a two-tailed Student's $t$-test to test if the multiple comparison corrected (Bonferroni correction) $p$-value was less than the critical value of 0.05. In the case of ethnicity we used one-way analysis of variance to compare the reconstruction accuracies of faces with different ethnicities.

We were able to reject the null hypothesis for the measures complexity, femininity and masculinity, but failed to do so for attractiveness, ethnicity and prototypicality. Specifically, we observed a significant negative correlation ($r = -0.3067$) between stimulus complexity and reconstruction accuracy. Furthermore, we found that masculinity and reconstruction accuracy were significantly positively correlated ($r = 0.3841$). Complementing this result, we found a negative correlation ($r = -0.3961$) between femininity and reconstruction accuracy. We found no effect of attractiveness, ethnicity and prototypicality on the quality of reconstructions. We then compared the complexity levels of the images of each gender and found that female face images were significantly more complex than male face images ($p < 0.05$, Student's $t$-test), pointing to complexity as the factor underlying the relationship between reconstruction accuracy and gender. This result demonstrates the importance of taking stimulus complexity into account while making inferences about factors driving the reconstructions from brain responses.

## 4 Conclusion

In this study we combined probabilistic inference with deep learning to derive a novel deep neural decoding approach. We tested our approach by reconstructing face stimuli from BOLD responses at an unprecedented level of accuracy and detail, matching the target stimuli in several key aspects such as gender, skin color and facial features as well as identifying perceptual factors contributing to the reconstruction accuracy. Deep decoding approaches such as the one developed here are expected to play an important role in the development of new neuroprosthetic devices that operate by reading subjective information from the human brain.

## Acknowledgments

This work has been partially supported by a VIDI grant (639.072.513) from the Netherlands Organization for Scientific Research and a GPU grant (GeForce Titan X) from the Nvidia Corporation.

## Footnotes

*We also experimented with the VGG-ImageNet pretrained model, which failed to match the reconstruction performance of the VGG-Face model, while their encoding performances were comparable in non-face related brain areas. We plan to further investigate other models in detail in future work.

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
