[Supplementary Material]

# Reconstructing perceived faces from brain activations with deep adversarial neural decoding (supplementary material)

**Yağmur Güçlütürk\*, Umut Güçlü\*,**
**Katja Seeliger, Sander Bosch,**
**Rob van Lier, Marcel van Gerven,**
Radboud University,
Donders Institute for Brain, Cognition and Behaviour
Nijmegen, the Netherlands
{y.gucluturk, u.guclu}@donders.ru.nl

## Encoding analysis

In addition to the decoding analysis reported in the main manuscript, an encoding analysis was performed to investigate how accurately the stimulus-evoked voxel responses in different brain areas in the visual cortex can be predicted as a linear combination of the latent features.

Twenty-six regions of interest (ROIs) in the visual cortex were defined based on the HCP MMP 1.0 parcellation [1]. These ROIs were projected from the HCP MMP 1.0 parcellation to the native functional volumetric space via the fsaverage surface space, native surface space and native anatomical volumetric space.

The analysis was performed on the average functional data of the two subjects. The data were averaged after further projecting each ROI from the native functional space of the individual subjects to a common functional space via hyperalignment [2].

A ridge regression model per voxel was estimated on the training set. The regularization coefficients of the models were selected with leave-one-out cross-validation. The models were evaluated on the test set. Model performance was defined as the Pearson correlation coefficient between the observed and predicted voxel responses.

Table 1 shows the summary statistics of the model performance for each ROI. Figure 1 shows the histograms of the model performance for each ROI. As expected, performance was higher ($\geq 0.10$) in certain downstream (V3CD, LO2, LO3, V4t), face-selective (PIT) and ventral (V8) areas. Interestingly, this was also the case in V1 (primary) and V3B (dorsal), but not in FFC (face-selective).

Table 1: Summary statistics (mean $\pm$ SEM (max)) of the model performance over individual voxels in each ROI.

| Category | | | | | | | | | |
|---|---|---|---|---|---|---|---|---|---|
| **Primary** | **V1** $0.10 \pm 0.00$ (0.61) | | | | | | | | |
| **Early** | **V2** $0.06 \pm 0.00$ (0.51) | **V3** $0.06 \pm 0.00$ (0.61) | **V4** $0.08 \pm 0.01$ (0.52) | | | | | | |
| **Dorsal** | **V3A** $0.08 \pm 0.01$ (0.54) | **V3B** $0.14 \pm 0.01$ (0.52) | **V6** $-0.03 \pm 0.01$ (0.39) | **V6A** $0.04 \pm 0.01$ (0.41) | **V7** $0.07 \pm 0.01$ (0.45) | **IPS1** $0.05 \pm 0.01$ (0.44) | | | |
| **Ventral** | **V8** $0.11 \pm 0.01$ (0.54) | **VVC** $0.05 \pm 0.01$ (0.47) | **PIT** $0.12 \pm 0.01$ (0.53) | **FFC** $0.06 \pm 0.01$ (0.52) | **VMV1** $0.01 \pm 0.01$ (0.36) | **VMV2** $0.04 \pm 0.01$ (0.40) | **VMV3** $0.07 \pm 0.01$ (0.38) | | |
| **MT+ & Nbr. Areas** | **V3CD** $0.10 \pm 0.01$ (0.49) | **LO1** $0.09 \pm 0.01$ (0.41) | **LO2** $0.11 \pm 0.01$ (0.42) | **LO3** $0.11 \pm 0.01$ (0.53) | **V4t** $0.10 \pm 0.01$ (0.53) | **FST** $0.06 \pm 0.01$ (0.50) | **MT** $0.07 \pm 0.01$ (0.46) | **MST** $0.07 \pm 0.01$ (0.39) | **PH** $0.05 \pm 0.01$ (0.43) |

Figure 1: Histograms of the model performance over individual voxels in each ROI.