[Reviews · NeurIPS 2017]

Reviewer 1



This paper presents a very interesting work of reconstructing a visual stimuli from brain activity, measured by fMRI, in a generative adversarial network (GAN) framework. Overall, the proposed idea is clearly described and the exhaustive experiments and analysis well supports the idea. However, there are still some points that should be clarified. - Eq. (5): The notations of ${\bf B}$ and $\beta_{-}$ are not defined. - In Eq. (6), from line 1 to line 2, to this reviewer’s understanding, it utilizes the relations of $y=B^{T}z$, $B^{-1}B=I$, and $B^{T}(B^{T})^{-1}=I$. However, as specified in the line 83 of the same page, $B\in\mathbb{R}^{p\times q}$, i.e., asymmetric. Hence, it’s necessary to explain how the relation between line 1 and line 2 in Eq. (6) holds. - In “fMRI dataset”, it is not clear what values from the deconvolve fMRI signals were used as the response $y$. What is the dimension of $y$? - Table 1: As reference, it would be informative to provide the upper limit of the three metrics obtained by the latent feature $z$ obtained by $\phi(x)$.

Reviewer 2



The authors propose a brain decoding model tailored to face reconstruction from BOLD fMRI measurements of perceived faces. There are some promising aspects to this contribution, but overall in its current state there are also a number of concerning issues. Positive points: - a GAN decoder was trained on face embeddings coming from a triplet loss or identity-predicting face embedding space to output the original images. Modulo my inability to follow the deluge of GAN papers closely, this is a novel contribution in that it is the application of the existant imagenet reconstruction GAN to faces. This itself may be on the level of a workshop contribution. - The reconstructed faces often coincide in gender and skin color with the original faces, indicating that the BOLD fMRI data contained this information. Reconstructing from BOLD fMRI is a very hard task and requires good data acquisition. The results indicate that a good job has been done on this front. - the feature reconstruction numbers indicate that using a GAN for face reconstruction is a step forward with respect to other (albeit much older and inherently weaker) methods. Points of concern: No part of the reconstruction method proposed seems novel (as claimed) and validation is shaky in parts. - The Bayesian framework seems superfluous. The reconstruction is performed using ridge regression with voxel importance weighted by inverse noise level. The subsection of deriving the posterior can be cut. [Notational quirk: If I understand correctly, B is not necessarily square nor necessarily invertible if it were. If the pseudoinverse is meant, it should be written with a different symbol, e.g. +] - No reason is given as to why PCA is performed on the face embeddings and how the number of components was chosen. (If the full embedding had been chosen, without PCA, would the reconstructions of embeddings look perfect? Is the application of PCA the "crucial aspect that made this work"? If so, please elaborate with experiments as to why this could be - projecting properly into the first two (semantically meaningful) axes does seem like an important thing to be able to do and could add robustness. - No information is provided as to which voxels were used for reconstruction. Were they selected according to how well they are predicted? If so, was the selection explicit or implicit by using the inverse noise variance weighting? If explicit (e.g. noise level cutoff), how many were selected and according to which cutoff criterion? Were these voxels mostly low-level visual or high-level visual? Could a restriction to different brain areas, e.g. known face-selective vs areas of the low-level visual hierarchy be used and evaluated? This question is not only neuroscientifically relevant. It would provide insight into how the decoding currently works. - The evaluation of the reconstruction seems to hang in the void a bit, as well as its comparison with other (incorrigibly worse) methods. The GAN that was trained outputs faces. It was trained to do this and it seems to do it well. This means that the baseline correlation of *any* output of the GAN will have a relatively high correlation with any face due to alignment of features. The permutation test of significance is appropriate and this outcome should be highlighted more (how about comparing to reconstructions from random vectors also?). A histogram of correlation scores in supplementary material would also provide insight. The comparison against the other methods seems a bit unfair. These other methods are (in modern view) clearly "all-purpose hacks", because there was not much else available to do at the time. Nobody expects eigenfaces to do well in any measure, nor does anybody expect mean and some restricted form of (diagonal?) covariance to be sufficient to model natural image data. These methods are thus clearly lacking in expressiveness and could never yield as good reconstructions as the neural network model. There are other face representations using active appearance modeling that could have been tried, which perform much better reconstruction. (No fMRI reconstruction has been attempted with these and thus it could be an interesting extra comparison.) This outcome is good for the vggface/GAN couple in terms of face representation and reconstruction (and as stated, this may be worth writing about separately if it hasn't been done before). It is however only the increase in feature decoding accuracy that shows that the representation is also slightly better for BOLD data (as opposed to reconstruction accuracy). However, looking at some of the eigenface/identity reconstructions, they also sometimes get core aspects of the faces right, so they are not doing that badly on certain perceptual axes. Lastly, why is it that the GAN reconstruction gets to have a "best possible reconstruction from this embedding" column, whereas eigenfaces and identity reconstruction do not get this privilege? Why does Table 1 not have the equivalent ratios for the other methods? - The title of this contribution is wildly inappropriate and must be changed to reflect the content of the work. It would be less unacceptable if there had also been a reconstruction from general natural images to parallel the face reconstruction, or if there had been several other restricted data types. In the current state the title must contain sufficient information to infer that it revolves around face reconstruction and nothing else. - Other possible axes of comparison: What about neural image representations trained on imagenet, such as Alexnet convolutional layers? How well does a GAN reconstruction from those do on faces? What about trying the face representation and reconstruction of the original DCGAN paper? All in all, the contribution is significant but could be much more so, if more experiments had been made.

Reviewer 3



Summary: The authors deploy an adverserial training process to improve decodability of faces from brain data. Reaction: In general, I don't have a lot to say about this paper, because it seems quite straightforward. It seems like a solid implementation of a nice idea. The explication is very clear, and the results are reasonably cleanly quantified. Obviously the results are imperfect (as can be seen from Table 1 and Fig 2) but they seem to be a clear improvement over other approaches. One minor confusion: the text describes the results of Table 1 as follows: "Table 1 shows three reconstruction accuracy metrics for both subjects in terms of the ratio of the reconstruction accuracy from the latent features to the reconstruction accuracy from brain responses." Isn't that reversed? Shouldn't it read: "Table 1 shows three reconstruction accuracy metrics for both subjects in terms of the ratio of the reconstruction accuracy from the brain responses to the reconstruction accuracy from latent features."? After all, the brain responses seem to construct significantly less good responses than the original latent features. I have a few concerns: (1) The set of faces used for testing seems rather limited -- eg. only faces from a single face-on angle, at a standard size and central position. How well would this same technique work on faces with more variation, e.g. at various rotations? If the technique become significantly less accurate, would that make the results less impressive? Also, it would interesting to see if the network could learn to produce the head-on image of a face seen at a non-standard angle. I understand that addressing this concern is probably not immediately possible, because of a limitation of the underlying neural dataset. (2) It's not clear that fig 3 or 4A add much to the overall story. I know that with variational latent models interpolations and feature tilings are often computed. Fig 3 shows that the model has the same apparent qualitative interpretability for latent feature dimensions as many other such models, and is not a surprise therefore. Fig 4A shows qualitatively the interpolation is smooth-ish both for the latent model and (kind of) for brain responses. I'm not sure what is really added by this about brain decodability by these analyses. How much "better" are these interpolations that simple pixel morphing? (And on what axes of betterness?) In general I find such interpolation illustrations hard to put into context (as to how "good" such interpolations are) even in "usual" pure machine-learning adversarial generation papers, and that doubly applies here. (3) Fig4B is more interesting, showing that reasonable and diverse samples of faces arise from sampling response priors of brain data. That's cool. I wish it was better quantified. In particular, I was some kind of quantitative analysis of the diversity and quality of these samples was developed, and then a comparison on such metrics made between results from their model and other approaches to brain decoding (e.g. the ones tested in Fig 5). I'd much rather see this comparison than the stuff in Fig 3 and 4A.